# Seroepidemiological survey and seropositivity rate for *Trypanosoma cruzi* infection in a community-based cardiac screening initiative in Feira de Santana, Bahia, Brazil

Felipe Silva Santos de Jesus[1,2]ᵒ, Isabella Moreira Gonzalez Fonseca[3]ᵒ, Ângelo Antônio Oliveira Silva[1,2], Noilson Lázaro Sousa Gonçalves[1], Daniel Dias Sampaio[2], Deborah Bittencourt Mothé[4], Paola Alejandra Fiorani Celedon[5], Dalila Luciola Zanette[5], Nilson Ivo Tonin Zanchin[6], Maria Carmo Pereira Nunes[3], Manoel Otávio da Costa Rocha[3], Craig Sable[7], Antonio Luiz Pinho Ribeiro[3], Fred Luciano Neves Santos [1,2,8]*

1 Advanced Health Public Laboratory, Gonçalo Moniz Institute, Oswaldo Cruz Foundation, Salvador, Bahia, Brazil, 2 Interdisciplinary Research Group in Biotechnology and Epidemiology of Infectious Diseases (GRUPIBE), Gonçalo Moniz Institute, Oswaldo Cruz Foundation, Salvador, Bahia, Brazil, 3 Medical School and University Hospital, Federal University from Minas Gerais, Belo Horizonte, Minas Gerais, Brazil, 4 Laboratory of Interaction Parasite Host and Epidemiology (LAIPHE), Gonçalo Moniz Institute, Oswaldo Cruz Foundation, Salvador, Bahia, Brazil, 5 Laboratory for Applied Science and Technology in Health, Carlos Chagas Institute, Oswaldo Cruz Foundation, Curitiba, Paraná, Brazil, 6 Structural Biology and Protein Engineering Laboratory, Carlos Chagas Institute, Oswaldo Cruz Foundation, Curitiba, Paraná, Brazil, 7 Ochsner Children's Hospital, New Orleans, Louisiana, United States of America, 8 Integrated Translational Program in Chagas disease from Fiocruz – Fio-Chagas, Rio de Janeiro, Rio de Janeiro, Brazil

ᵒ These authors contributed equally to this work.
* fred.santos@fiocruz.br

## Abstract

Chagas disease (CD), caused by *Trypanosoma cruzi*, is a significant public health issue in Latin America, particularly in endemic regions. This study integrates a seroepidemiological survey with large-scale echocardiographic screening conducted in Feira de Santana, a highly endemic city in Bahia, Brazil, to estimate the seropositivity rate of *T. cruzi* infection and identify associated risk factors. Peripheral blood samples were analyzed using in-house ELISA based on IBMP chimeric antigens and an indirect hemagglutination assay. Among 1,115 participants enrolled in the cardiac screening initiative, 140 underwent serological testing comprising individuals who screened positive based on clinical data, conventional ECG, and ECG-AI, and controls matched in a 2:1 ratio. Of these, 8.5% tested seropositive, with household exposure to triatomines identified as the strongest risk factor (prevalence ratio = 4.38, p = 0.004). Most seropositive individuals were migrants from other endemic areas, underscoring the influence of population mobility on CD epidemiology. This study highlights the importance of integrating diagnostic tools and vector control strategies into community-based health initiatives to improve early detection, reduce disease burden, and inform public health interventions in underserved regions.

**Data availability statement:** All relevant data are in the manuscript and its supporting information files.

**Funding:** This study was supported by the Edwards Lifesciences Foundation and the Coordination for the Improvement of Higher Education Personnel in Brazil (CAPES; Finance Code 001). NITZ and FLNS are research fellows of the National Council for Scientific and Technological Development-Brazil (CNPq; processes 304894/2023-0 and 306448/2023-8, respectively). The funders had no role in study design, data collection and analysis, decision to publish, or preparation of the manuscript.

**Competing interests:** The authors have declared that no competing interests exist.

## Author summary

Chagas disease, caused by the parasite *Trypanosoma cruzi*, remains a major health concern in Latin America, especially in areas with limited access to healthcare. In this study, we combined artificial intelligence-based electrocardiogram analysis and cardiac screening with serological testing to detect undiagnosed *T. cruzi* infections in Feira de Santana, a city in the state of Bahia, Brazil. By analyzing electrocardiogram data, epidemiological information, and laboratory assays, we identified individuals at increased risk and confirmed infection using validated serological methods. Our results showed that people who reported the presence of triatomine bugs inside their homes were significantly more likely to be infected, emphasizing the role of vector exposure in *T. cruzi* transmission. In addition, most seropositive individuals were unaware of their condition and had migrated from other endemic regions. These findings highlight the importance of combining innovative technologies with community-based screening initiatives can improve early detection of *T. cruzi* infection and guide public health strategies in vulnerable populations.

## Introduction

Chagas disease (CD), caused by the protozoan *Trypanosoma cruzi*, remains a significant public health issue in Latin America, affecting millions and contributing to considerable morbidity and mortality [1,2]. Despite advances in control measures, gaps in early detection and treatment persist, particularly in underserved regions [3]. Chronic CD can lead to cardiac involvement, which represents the leading cause of mortality, highlighting the need for innovative strategies to improve healthcare delivery and patient outcomes.

The state of Bahia is recognized as one of the Brazilian states with persistent endemicity for *T. cruzi* infection, where both vectorial and oral transmission routes have been historically documented. Recent seroepidemiological surveys continue to report significant seroprevalence rates in rural and peri-urban communities [4–6]. Bahia also harbors the greatest diversity of triatomine species in Brazil, including several species that have adapted to human dwellings, thereby sustaining the risk of domestic transmission [7–10]. New triatomine species have been described or revalidated in the state, and changes in vector ecology and human–environment interactions have posed additional challenges to control strategies.

Within Bahia, Feira de Santana is one of the most populous municipalities and serves as an important regional commercial and transportation hub. The city has reported substantial morbidity and mortality associated with CD, reflecting the challenges in vector control, diagnosis, and clinical management [11]. Marked socioeconomic inequalities, population mobility, and limited access to specialized healthcare services further contribute to delays in the identification and follow-up of individuals infected with *T. cruzi*. In this context, innovative approaches that integrate diagnostic

technologies into existing healthcare initiatives are essential to improve case detection and reduce disease burden. International initiatives such as the *Every Heartbeat Matters Program* have emerged as promising strategies to address cardiovascular health disparities in resource-limited settings [12]. This initiative incorporates telehealth-enabled electro-cardiography (teleECG), artificial intelligence–based ECG analysis, and targeted echocardiographic screening to improve the detection and management of structural heart disease at the community level. Given the strong association between chronic *T. cruzi* infection and cardiac involvement, such platforms offer an opportunity to integrate seroepidemiological investigations into large-scale cardiac screening programs.

In this study, we leveraged data from a seroepidemiological survey conducted alongside the *Every Heartbeat Matters* initiative in Feira de Santana. The primary objectives were to estimate the seropositivity rate of *T. cruzi* infection, iden-tify epidemiological and environmental factors associated with infection, and assess the feasibility of integrating *T. cruzi* screening into community-based echocardiographic programs in endemic settings. The findings aim to inform public health strategies and strengthen approaches for the early detection and management of *T. cruzi* infection in high-burden regions.

## Methods

### Ethics

This study adhered to the principles of the Declaration of Helsinki and was approved by the Institutional Review Board (IRB) of the Gonçalo Moniz Institute, Oswaldo Cruz Foundation (IGM-FIOCRUZ), Salvador, Bahia, Brazil (protocol no. 67809417.0.0000.0040). Written informed consent was obtained from all participants prior to enrollment.

### Study population

This cross-sectional study was conducted in Feira de Santana, Bahia, Brazil, the second largest city in the state, located in the semi-arid region of Northeastern Brazil (11° 11′ S, 38° 58′ W) at an altitude of 234 m. According to the Brazilian Institute of Geography and Statistics (IBGE), the municipality has approximately 616,272 inhabitants and serves as an important commercial and transport hub [13]. The urban area is characterized by high population density and pronounced socioeconomic inequalities, particularly in peripheral neighborhoods with limited access to healthcare, sanitation, and adequate housing.

A total of 1,115 adults were recruited during a large-scale community-based cardiac screening initiative. All participants completed a structured questionnaire collecting sociodemographic data (age, sex, education, employment, household income), environmental and housing characteristics (construction materials, presence of domestic animals or chicken coops, and history of triatomine sightings), migration background, family history of Chagas disease, and previous medical diagnoses. Cardiovascular symptoms, including reduced exercise tolerance, chest pain, syncope, and palpitations, were also assessed. Questionnaires were administered by trained healthcare professionals or supervised medical students at the time of echocardiographic examination.

### Cardiac screening and risk stratification

All participants underwent a standard 12-lead electrocardiogram (ECG) acquired using a PC-based system (TEB, São Paulo, Brazil) and transmitted electronically to a telehealth platform for interpretation by cardiologists. ECG abnormalities were classified as major or minor according to the Minnesota Code criteria [14].

Transthoracic echocardiography was performed by a team of eight cardiologists using commercially available systems (Vivid Q, GE Healthcare; Affiniti 70, Philips, USA). Images were stored digitally and interpreted immediately after acquisi-tion; uncertain findings were resolved by consensus reading. Abnormal echocardiograms were defined by the presence of major structural or functional abnormalities, as previously described [12].

An ECG-based artificial intelligence (AI) algorithm, implemented as a deep neural network trained on standard 12-lead digital ECG waveforms (time–voltage signals), was applied to estimate the probability of *T. cruzi* infection [15]. Risk stratification combined the AI-ECG output with responses to three epidemiological questions assessing whether participants had: (i) a family member diagnosed with CD, (ii) lived in areas with known triatomine (kissing bug) exposure, and (iii) resided in wooden houses, a recognized risk factor for vector transmission.

Participants classified as at increased risk were invited to undergo serological testing. For comparison, control individuals classified as low risk were randomly selected using a computer-generated algorithm at a 2:1 ratio (controls to cases) and also underwent serological testing.

## Specimen collection and storage

Peripheral blood samples (9 mL) were collected into clot-activator tubes with gel separators (Vacuplast, Brazil). After clotting at room temperature for 30 minutes, samples were centrifuged at $3,000 \times g$ for 10 minutes. Serum aliquots were transported to the Gonçalo Moniz Institute (Oswaldo Cruz Foundation, Bahia) and stored at −20 °C until analysis.

## Serological testing

Anti-*T. cruzi* IgG antibodies were initially detected using in-house ELISA assays based on four chimeric recombinant antigens (IBMP-8.1, IBMP-8.2, IBMP-8.3, and IBMP-8.4), following established protocols [16]. Recombinant antigens were expressed in *Escherichia coli* BL21-Star (DE3), purified by affinity and ion-exchange chromatography, and quantified using a Qubit 2.0 fluorometer (Invitrogen, Carlsbad, CA, USA). ELISAs were performed in 96-well microplates (Greiner Bio-One, Austria). Plates were sensitized with the antigens and blocked with a solution of Well-Champion (Ken-Em-Tec Diagnostics A/S, Denmark). Serum samples were diluted 1:100 in PBS-T (PBS containing 0.05% Tween-20; pH 7.4) and incubated at 37 °C for 30 minutes. After washing with PBS-T, a goat anti-human IgG-HRP conjugate (Bio-Manguinhos, FIOCRUZ, Brazil) was added and incubated at 37 °C for 30 minutes. Following additional washing, immune complexes were identified using TBM substrate (Kem-En-Tec, Denmark) with a 10-minute incubation in the dark at room temperature. The reaction was stopped with 0.3 M $H_2SO_4$, and optical density (OD) was measured at 450 nm using a SPECTRAmax 340PC microplate reader (Molecular Devices, USA).

All samples were additionally tested using a commercial indirect hemagglutination assay (Imuno-HAI Chagas, WAMA Diagnóstica, Brazil), performed according to the manufacturer's instructions. Sera were diluted 1:32 in assay diluent supplemented with 2-mercaptoethanol to reduce nonspecific reactivity. Qualitative results were interpreted visually, with reactive samples formed a compact erythrocyte button at the bottom of the well, whereas reactive samples producing diffuse erythrocyte mats and non-reactive samples forming compact buttons. Reactive samples were further titrated by serial two-fold dilutions, with titers ≥1:32 considered positive.

## Reference testing

In the absence of a single gold standard for detecting anti-*T. cruzi* antibodies, infection status was determined using a combined approach based on indirect hemagglutination results and latent class analysis (LCA) [17–19]. Within the LCA framework, seropositivity was defined as reactivity in at least two of the four IBMP ELISA, yielding posterior probabilities between 87.9% and 100%; while seronegativity required non-reactivity in at least three assays, with posterior probabilities ≤0.8%. Concordance between LCA and indirect hemagglutination was assessed for all samples. All samples classified as seropositive or presenting discordant results were further evaluated using two commercial ELISA assays, Gold ELISA Chagas (REM Indústria e Comércio Ltda, Brazil) and Bioelisa Chagas Recombinante (Bioclin Quibasa Química Básica, Brazil), to provide additional confirmation.

## Data analysis

Descriptive statistics summarized sociodemographic and epidemiological characteristics. Age was expressed as median and interquartile range (IQR), and categorical variables (e.g., sex, serological test results) as relative frequencies. Associations between *T. cruzi* seropositivity and potential risk factors, including prior exposure to triatomines, contact with kissing bugs, presence of a chicken coop, and familial history of CD, were evaluated using contingency tables. Seropositivity rates were calculated as the proportion of seropositive cases among exposed individuals relative to non-exposed individuals, with 95% confidence intervals (CI) and p-values derived from Fisher's exact test. Statistical significance was set at p-value < 0.05. Additionally, prevalence ratio (PR) was performed to assess the association of sociodemographic and environmental factors with positivity. Multivariable logistic regression models integrating ECG findings and epidemiological variables were used to estimate the probability of *T. cruzi* infection.

## Results

Between August 24 and 27, 2024, a total of 1,115 individuals participated in the *Every Heartbeat Matters* cardiac screening initiative conducted at the Holy House of Mercy in Feira da Santana (Fig 1). In the overall screened population, the median age was 57 years (IQR: 44–64 years), and women accounted for 78.4% of participants, corresponding to a

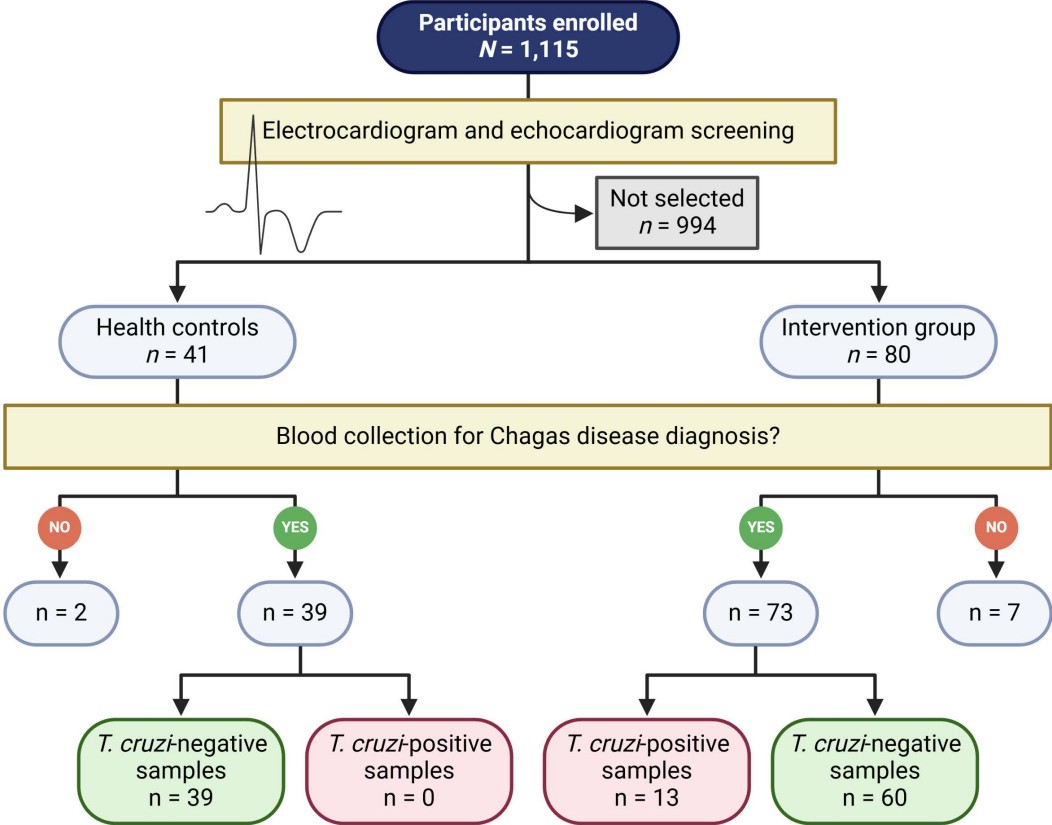

**Fig 1. Study design of a seroepidemiological survey for Chagas disease in Feira de Santana, Bahia, Brazil.** The figure summarizes participant selection within the Every Heartbeat Matters cardiac screening initiative, with ECG and echocardiographic evaluation, risk stratification, and subsequent serological testing using IBMP chimeric antigen–based ELISA and commercial assays. The design follows the Standards for Reporting of Diagnostic Accuracy Studies (STARD) guidelines.

female-to-male ratio of 4.9:1. According to the Minnesota Code criteria [14], 284 individuals (27.5%) presented minor ECG abnormalities, while 118 (11.4%) showed major abnormalities.

Based on the combined risk stratification approach integrating epidemiological information, conventional ECG findings, and ECG-AI output, 121 individuals were selected for serological evaluation. Of these, 80 were classified as being at increased risk for *T. cruzi* infection, and 41 low-risk individuals were randomly selected as controls at a 2:1 ratio. Blood samples were obtained from 112 participants, while the remaining screened individuals (n = 994) were not included in serological testing (S1 Table).

Among the 73 participants with ECG abnormalities who underwent serological testing, 13 (17.8%) were classified as seropositive for *T. cruzi*. In contrast, none of the 39 control individuals tested positive, resulting in a seropositivity rate of 0% in this group. Overall, the seropositivity rate among all tested participants was 11.6% (13/112) (Fig 1).

False-positive alerts generated by the ECG-AI algorithm were observed in 22 individuals. Among these, six presented major ECG abnormalities and 16 presented minor abnormalities; however, all were classified as seronegative by both indirect hemagglutination and latent class analysis. Echocardiographic screening among individuals with confirmed *T. cruzi* infection revealed left or right ventricular involvement in only two participants, while the remaining seropositive individuals showed no major structural abnormalities at the time of screening.

## Serological classification and assay concordance

The indirect hemagglutination assay classified 99 samples (88.4%) as non-reactive and 13 samples (11.6%) as reactive for anti-*T. cruzi* antibodies. Latent class analysis yielded an identical classification, with all 99 seronegative samples showing no reactivity to any of the four IBMP antigens and posterior probabilities of zero for positivity (S1 Table). Among the 13 seropositive samples, eight (61.5%) reacted with all four IBMP antigens, two (15.4%) reacted with three antigens, and three (23.1%) reacted with two antigens. Posterior probabilities for positivity ranged from 96.9% to 100%, confirming the classification of *T. cruzi* infection. No discordant results between indirect hemagglutination and LCA were observed. In addition, both commercial ELISA assays (Gold ELISA Chagas and Bioelisa Chagas Recombinante) detected seropositivity in 100% (13/13) of the confirmed positive samples (Table 1).

**Table 1. Reactivity patterns of serum samples from participants screened for Chagas disease in Feira de Santana, Bahia, Brazil.**

| Participant number | In-house assays with IBMP antigens | | | | | | Commercial kits | | |
|---|---|---|---|---|---|---|---|---|---|
| | -8.1 | -8.2 | -8.3 | -8.4 | LCA | P (%) * | IHA[1] | GEC[2] | BIO[3] |
| 1 | 2.16 | 0.55 | 1.07 | 0.41 | POS | 96.9 | 1:64 | 2.64 | 3.96 |
| 2 | 2.00 | 0.59 | 1.03 | 0.58 | POS | 96.9 | 1:64 | 2.54 | 3.53 |
| 3 | 2.04 | 1.58 | 1.83 | 2.17 | POS | 100 | 1:128 | 12.41 | 3.03 |
| 4 | 0.75 | 1.47 | 1.53 | 1.54 | POS | 100 | 1:256 | 7.35 | 3.22 |
| 5 | 2.75 | 1.84 | 1.72 | 1.85 | POS | 100 | 1:256 | 9.54 | 2.57 |
| 6 | 2.95 | 1.83 | 1.69 | 2.47 | POS | 100 | 1:64 | 7.89 | 3.15 |
| 7 | 2.41 | 1.54 | 1.85 | 1.49 | POS | 100 | 1:256 | 11.01 | 2.99 |
| 8 | 1.44 | 0.62 | 0.58 | 1.01 | POS | 99 | 1:64 | 2.47 | 2.68 |
| 9 | 1.95 | 1.17 | 0.76 | 1.23 | POS | 100 | 1:128 | 5.27 | 2.29 |
| 10 | 1.89 | 1.77 | 1.82 | 2.48 | POS | 100 | 1:512 | 12.71 | 3.57 |
| 11 | 2.84 | 1.08 | 1.90 | 2.03 | POS | 100 | 1:256 | 10.56 | 3.26 |
| 12 | 2.56 | 1.52 | 1.81 | 1.73 | POS | 100 | 1:64 | 10.66 | 2.33 |
| 13 | 2.24 | 1.16 | 1.44 | 1.75 | POS | 100 | 1:512 | 9.01 | 2.61 |

[1]IHA (Imuno-HAI Chagas); [2]GEC (Gold ELISA Chagas); [3]BIO (Bioelisa Chagas Recombinante); LCA (Latent Class Analysis); POS (Positive). *Probability of being *T. cruzi*-positive by LCA.

## Demographic and epidemiological characteristics of seropositive individuals

Among individuals classified as seropositive for *T. cruzi*, the median age was 60 years (IQR: 39–69), and the female-to-male ratio was 5.5:1. Seronegative individuals had a median age of 56 years (IQR: 44–64) and a female-to-male ratio of 4.8:1 (S1 Table). Notably, 69.2% of seropositive participants were not born in Feira de Santana but had migrated from other endemic municipalities within the state of Bahia, including Conceição do Jacuípe (n = 1), Coração de Maria (n = 1), Ipirá (n = 1), Mundo Novo (n = 1), Muniz Ferreira (n = 1), Santa Maria da Vitória (n = 1), Utinga (n = 1), and Santo Estevão (n = 2) (Fig 2). This suggests migration as a potential factor influencing local epidemiological trends.

Although most seropositive individuals reported awareness of their clinical condition, 23% were unaware of their *T. cruzi* infection status prior to participation in the screening initiative (Fig 2).

All seropositive individuals were older than 46 years and were either retired or beneficiaries of government assistance programs (Fig 2), except for one family cluster. This family consisted of two adults and three children and epidemiologically likely to a suspected episode of oral transmission after consumption of açaí juice in February 2024 near Feira de Santana. Two children (aged 9 and 13 years) and both parents (aged 40 and 38 years) tested seropositive, while one child who did not consume the juice tested seronegative. All infected family members received etiological treatment and remained under clinical follow-up.

## Risk factor analysis

In univariate analysis, self-reported observation of triatomine bugs inside the household was associated with a higher prevalence of *T. cruzi* seropositivity (PR = 4.38; 95% CI: 1.44–13.3, p = 0.004). Because triatomine exposure was not systematically compared between seropositive and seronegative individuals in the entire screened population, this association should be interpreted cautiously, as it may reflect broader characteristic of the surveyed community rather than a causal relationship. Other evaluated variables, including reported knowledge of the vector, presence of a chicken coop near the household, and family history of CD, were not significantly associated with seropositivity. This suggests that while reporting triatomine presence indoors may indicate increased risk, the association requires confirmation in broader community-based analyses (Table 2).

## Discussion

This study leverages a community-based cardiac screening initiative to conduct a seroepidemiological survey, integrating large-scale ECG and echocardiographic screening for the identification of CD. A significant association was identified between self-reported household observation of triatomine bugs and *T. cruzi* seropositivity, underscoring vector exposure as a key factor associated with CD occurrence in endemic regions. While other environmental factors, such as the presence of chicken coops near households and family history of CD, were not significantly associated, direct exposure to the vector emerged as the strongest factor linked to a higher prevalence of infection. These findings are consistent with previous studies emphasizing the critical role of domestic vector infestation in sustaining transmission cycles [20,21].

The overall seropositivity rate of *T. cruzi* infection observed in this study (8.5%) is comparable to reports from other endemic areas of Bahia [4–6], reinforcing the persistent public health challenge posed by CD. The high seropositivity rate among individuals reporting the presence of triatomines inside their households highlights the need for effective vector control programs. In addition to reducing domestic infestation, community education focused on vector recognition and preventive measures remains essential to disrupt transmission dynamics.

A notable observation was that most *T. cruzi*-positive individuals currently residing in Feira de Santana were born in other endemic areas. This finding suggests that many infections likely occurred prior to migration, highlighting the influence of population mobility on CD epidemiology. However, it remains unclear whether migration itself represents an independent risk factor for *T. cruzi* transmission or whether expose occurred in the individuals' places of origin before

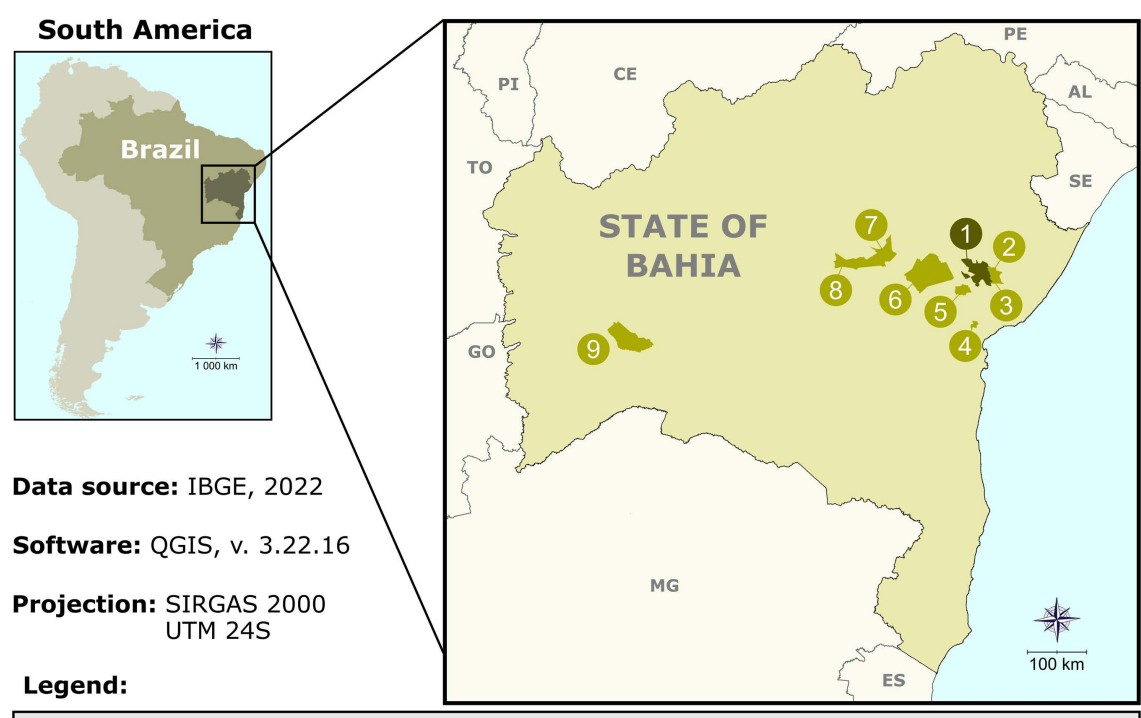

**Data source:** IBGE, 2022

**Software:** QGIS, v. 3.22.16

**Projection:** SIRGAS 2000
UTM 24S

**Legend:**

| | Birthplace | Participant number | Sex | Age (in years) | Occupation | Previous CD awareness |
|---|---|---|---|---|---|---|
| 1 | Feira de Santana | 1 | Male | 9 | Student | Yes |
| | | 2 | Female | 13 | Student | Yes |
| | | 3 | Female | 73 | Retired | Yes |
| | | 4 | Female | 64 | Retired | No |
| 2 | Coração de Maria | 5 | Female | 61 | Washerwoman | Yes |
| 3 | Conceição do Jacuípe | 6 | Female | 77 | Retired | No |
| 4 | Muniz Ferreira | 7 | Female | 73 | Retired | Yes |
| 5 | Santo Estevão | 8 | Male | 40 | Unemployed | Yes |
| | | 9 | Female | 38 | Housewife | Yes |
| 6 | Ipirá | 10 | Female | 65 | Retired | Yes |
| 7 | Mundo Novo | 11 | Female | 59 | Retired | Yes |
| 8 | Utinga | 12 | Female | 60 | Housewife | No |
| 9 | Santa Maria da Vitória | 13 | Female | 46 | Welfare recipient | Yes |

**Fig 2. Geographic origins of individuals seropositive for *Trypanosoma cruzi* in Feira de Santana, Bahia, Brazil.** The map shows the municipalities of birth of participants who tested positive for Chagas disease during the Every Heartbeat Matters screening initiative. Public domain digital maps were obtained from the Brazilian Institute of Geography and Statistics (IBGE) and analyzed using QGIS version 3.10 (Geographic Information System, Open-Source Geospatial Foundation Project. http://qgis.osgeo.org).

**Table 2. Univariate analysis of epidemiological factors associated with *Trypanosoma cruzi* seropositivity in participants from Feira de Santana, Bahia, Brazil.**

| Variables | Univariate analysis | | |
|---|---|---|---|
| | PR | 95%CI | p-value |
| Prior exposure to triatomines | 4.38 | 1.44 to 13.3 | 0.004 |
| Presence of a chicken coop | 0.82 | 0.27 to 2.48 | 0.730 |
| Familial history of CD | 1.81 | 0.59 to 5.55 | 0.285 |

PR (Prevalence ratio), β (Log-odds coefficient), CD (Chagas disease) CI (Confidence interval).

relocation. The absence of longitudinal data on the timing of infection limits causal inferences related to migration, as transmission may have occurred either before or after resettlement. Furthermore, differences in vector control programs and access to healthcare across endemic regions may differentially influence infection risks among migrant populations. These findings support the need for targeted interventions and emphasize the importance of integrating migrant populations into local healthcare systems. Tailored strategies to improve access to diagnostic and treatment services for migrant communities are essential to reduce the overall burden of CD in endemic regions.

Further analysis revealed that individuals who reported finding triatomines in their households presented fourfold higher prevalence of *T. cruzi* infection (PR = 4.38, p = 0.004) compared with those who did not report vector presence indoors. This result further underscores the central role of domestic vector infestation in maintaining transmission cycles and corroborates previous research identifying in-home triatomine presence as a critical determinant of infection risk [20,21]. The lack of significant associations for other variables, such as general knowledge about the vector or the presence of chicken coops, reinforces the primacy of direct vector contact in transmission dynamics.

From a methodological perspective, the use of IBMP chimeric antigens for serological screening, complemented by commercial assays for case validation, resulted in high diagnostic accuracy. The application of latent class analysis to classify seropositive cases minimized misclassification bias and enhanced the reliability of the findings. These methodological strengths demonstrate the value of integrating innovative diagnostic tools into community-based health initiatives to effectively address CD in endemic regions.

Although most participants were aware of their clinical conditions, it is noteworthy that 23% of individuals identified as *T. cruzi*-positive were unaware of their infection status. This finding highlights the insidious nature of chronic CD, which may remain asymptomatic or manifest with nonspecific symptoms for prolonged periods, thereby complicating timely diagnosis. Limited access to healthcare services, suboptimal screening practices, and low disease awareness in at-risk communities likely contribute to this diagnostic gap. The fact that nearly one quarter of seropositive individuals were previously undiagnosed underscores the need for enhanced community-based screening programs, particularly in endemic settings. By integrating serological testing with cardiac screening initiatives, as demonstrated in this study, early detection and intervention become feasible, potentially reducing disease progression and long-term morbidity. This observation aligns with previous studies reporting substantial proportions of undiagnosed infections in similar contexts [22–24].

However, the cross-sectional design of this study limits causal inferences, and the relatively small number of seropositive cases may affect the generalizability of the findings. In addition, recruitment through a cardiac screening initiative introduces potential selection bias, as individuals with underlying cardiovascular abnormalities may not be representative of the broader population at risk for *T. cruzi* infection. This could lead to an overestimation of seropositivity in this subgroup while underrepresenting asymptomatic or early-stage infections in the general community. To mitigate sampling bias in serologic testing, a computer-generated random selection of low-risk controls based on ECG-AI scores was applied, improving the representativeness of the tested sample. Future longitudinal studies with larger, community-based

populations, such as those planned within the Oxente Chagas Project [25], are needed to further explore the temporal dynamics of infection risk and validate these findings.

In conclusion, this study provides important insights into the epidemiology of *T. cruzi* infection in Feira de Santana, Bahia, emphasizing the central role of vector exposure in CD transmission. The integration of seroepidemiological surveys with cardiac screening initiatives represents a valuable model for identifying at-risk populations and improving early detection of CD. Continued investment in vector control, diagnostic innovation, and public health education remains essential to reduce the burden of CD in endemic regions [20,26]. These findings underscore the importance of community-based health initiatives in advancing CD control and improving outcomes in underserved and vulnerable populations.

## Supporting information

**S1 Table. Sociodemographic, epidemiological, and serological characteristics of the study samples.**
(XLSX)

## Acknowledgments

We are grateful to the Holy House of Mercy team in Feira da Santana for their invaluable assistance in performing this study.

## Author contributions

**Conceptualization:** Isabella Moreira Gonzalez Fonseca, Maria Carmo Pereira Nunes, Manoel Otávio da Costa Rocha, Craig Sable, Antonio Luiz Pinho Ribeiro.

**Data curation:** Nilson Ivo Tonin Zanchin, Maria Carmo Pereira Nunes, Craig Sable, Antonio Luiz Pinho Ribeiro, Fred Luciano Neves Santos.

**Formal analysis:** Felipe Silva Santos de Jesus, Isabella Moreira Gonzalez Fonseca, Ângelo Antônio Oliveira Silva, Daniel Dias Sampaio, Deborah Bittencourt Mothé, Paola Alejandra Fiorani Celedon, Nilson Ivo Tonin Zanchin, Maria Carmo Pereira Nunes, Craig Sable, Antonio Luiz Pinho Ribeiro, Fred Luciano Neves Santos.

**Funding acquisition:** Nilson Ivo Tonin Zanchin, Maria Carmo Pereira Nunes, Manoel Otávio da Costa Rocha, Craig Sable, Antonio Luiz Pinho Ribeiro, Fred Luciano Neves Santos.

**Investigation:** Isabella Moreira Gonzalez Fonseca, Ângelo Antônio Oliveira Silva, Paola Alejandra Fiorani Celedon, Nilson Ivo Tonin Zanchin, Maria Carmo Pereira Nunes, Craig Sable, Antonio Luiz Pinho Ribeiro, Fred Luciano Neves Santos.

**Methodology:** Felipe Silva Santos de Jesus, Isabella Moreira Gonzalez Fonseca, Ângelo Antônio Oliveira Silva, Noilson Lázaro Sousa Gonçalves, Paola Alejandra Fiorani Celedon, Dalila Luciola Zanette, Nilson Ivo Tonin Zanchin, Maria Carmo Pereira Nunes, Craig Sable, Antonio Luiz Pinho Ribeiro, Fred Luciano Neves Santos.

**Project administration:** Maria Carmo Pereira Nunes, Manoel Otávio da Costa Rocha, Craig Sable, Antonio Luiz Pinho Ribeiro, Fred Luciano Neves Santos.

**Resources:** Craig Sable.

**Software:** Isabella Moreira Gonzalez Fonseca, Craig Sable.

**Supervision:** Nilson Ivo Tonin Zanchin, Maria Carmo Pereira Nunes, Manoel Otávio da Costa Rocha, Craig Sable, Antonio Luiz Pinho Ribeiro.

**Validation:** Nilson Ivo Tonin Zanchin, Maria Carmo Pereira Nunes.

**Writing – original draft:** Felipe Silva Santos de Jesus, Isabella Moreira Gonzalez Fonseca, Maria Carmo Pereira Nunes, Fred Luciano Neves Santos.

**Writing – review & editing:** Felipe Silva Santos de Jesus, Isabella Moreira Gonzalez Fonseca, Ângelo Antônio Oliveira Silva, Noilson Lázaro Sousa Gonçalves, Daniel Dias Sampaio, Deborah Bittencourt Mothé, Paola Alejandra Fiorani Celedon, Dalila Luciola Zanette, Maria Carmo Pereira Nunes, Manoel Otávio da Costa Rocha, Craig Sable, Antonio Luiz Pinho Ribeiro, Fred Luciano Neves Santos.

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
