## [Decision Letter · Decision Letter 0]

31 Jul 2025

Seroepidemiological survey and seropositivity rate for Trypanosoma cruzi infection in a community-based cardiac screening initiative in Feira de Santana, Bahia, Brazil

Dear Dr. Santos,

Thank you for submitting your manuscript to PLOS Neglected Tropical Diseases. After careful consideration, we feel that it has merit but does not fully meet PLOS Neglected Tropical Diseases's publication criteria as it currently stands. Therefore, we invite you to submit a revised version of the manuscript that addresses the points raised during the review process.

Please submit your revised manuscript within 60 days Sep 29 2025 11:59PM. If you will need more time than this to complete your revisions, please reply to this message or contact the journal office at plosntds@plos.org. Please include the following items when submitting your revised manuscript:

We look forward to receiving your revised manuscript.

Kind regards,

Marilia Sá Carvalho

Academic Editor

Guilherme Werneck

Section Editor

Shaden Kamhawi

co-Editor-in-Chief

Paul Brindley

co-Editor-in-Chief

**Journal Requirements:**

- We do not publish any copyright or trademark symbols that usually accompany proprietary names, eg ©,  ®, or TM  (e.g. next to drug or reagent names). Therefore please remove all instances of trademark/copyright symbols throughout the text, including:

- ® on pages: 7, 9, and 10.

**Reviewers' Comments:**

Reviewer's Responses to Questions

**Key Review Criteria Required for Acceptance?**

**Methods:**

-Are the objectives of the study clearly articulated with a clear testable hypothesis stated?

-Is the study design appropriate to address the stated objectives?

-Is the population clearly described and appropriate for the hypothesis being tested?

-Is the sample size sufficient to ensure adequate power to address the hypothesis being tested?

-Were correct statistical analysis used to support conclusions?

-Are there concerns about ethical or regulatory requirements being met?

Reviewer #1: Once Bahia State presents one of the highest triatomine infestation rates in Brazil, the hypothesis of the study should be presented in a testable way.

The design is appropriate and in accordance with the objectives and the obtained results; the sample size is appropriate and the ethical issues are well addressed.

Reviewer #2: The objectives are clear and consistent with the cross-sectional study design. I believe that the sample size is sufficient and appropriate for the hypothesis tested using algorithm-based artificial intelligence as a technological tool, and the statistical analysis supports the conclusions. There are no ethical concerns.

Reviewer #3: In the absence of a parasitological test, the WHO recommends and is internationally accepted that a definitive serological diagnosis must be based on two distinct serological tests. This is especially important in trypanosomatids, where cross-reactivity with other Trypanosoma species and even Leishmania is reported. The inclusion of a second test should not be presented as “Reference Testing”, but as second confirmatory for the ELISA. Even using four chimeric proteins, it remains the same test (ELISA), and the high chance of cross-reactivity (Leishmaniasis is a reality throughout Brazil) increases the risk of false positives. Morevoer, The echocardiographic test cannot be considered a screening test, given that several other diseases present with cardiac dysfunction, and a high percentage of Chagas seropositive individuals do not develop symptoms.

The “structured questionnaire” (line 100) wasn't presented, and I imagine it's not just the three questions that appear in the following paragraph.

Electrocardiogram (ECG) and echocardiography: Which equipment? How was it performed? By the same professional? Who interpreted the results? The methodological approach is lacking in detail.

Artificial intelligence (AI)-based algorithm: How was it done? What was the methodology?

“Clinical data” (line 110): What clinical symptoms formed the basis for the clinical data? Who collected it? What aspects were assessed?

**Results:**

-Does the analysis presented match the analysis plan?

-Are the results clearly and completely presented?

-Are the figures (Tables, Images) of sufficient quality for clarity?

Reviewer #1: The results are well presented and ilustrated in the tables and figs

The legends must be revised

Reviewer #2: The results are presented as planned, in a clear and concise manner with self-explanatory figures and tables. Positive highlights of the study include the integration of new diagnostic tools into health initiatives and the efficient approach to detecting Chagas disease in endemic regions.

Reviewer #3: The association with presence of triatomine bugs cannot be done because authors didn’t compare this trait to the negatives. It could just reflect the overall population characteristic.

Lines 177-178: These results need to be better described. If they are part of another study, you need to reference the other study that contains these data.

The “AI algorithm” (line 179): There is also a lack of information on how it was produced and tested.

“…based on predefined criteria…” (line 181): criteria not presented in the article

Line 187: Why false positives, since several other diseases besides Chagas' disease cause heart problems? Several other diseases present with cardiac dysfunction, and a high percentage of Chagas seropositive individuals do not develop symptoms.

Lines 202-203: This affirmation cannot be done because negative samples, in one and/or the commercial tests, were evaluated.

**Conclusions:**

-Are the conclusions supported by the data presented?

-Are the limitations of analysis clearly described?

-Do the authors discuss how these data can be helpful to advance our understanding of the topic under study?

-Is public health relevance addressed?

Reviewer #1: Conclusions are in accordance with the obtained results which are well discussed

The manuscript presents a relevant analysis on Chagas disease, which represents an important public health concern

Reviewer #2: The conclusions are appropriate to the data and highlight the limitations of the study. There is good discussion of the data in light of the epidemiologic indicators related to Chagas disease. It also addresses the importance of reporting suspected cases in primary care and public health services, early diagnosis, and gaps in the availability of etiologic treatment as relevant factors in controlling parasite transmission. It also highlights the importance of integrating seroepidemiologic surveys with cardiac screening, which I believe is a valuable model for identifying at-risk populations and improving early detection of T. cruzi infection. The data also show that it is necessary and essential to invest in diagnostic innovation to fill the gaps that still exist in the availability of etiologic treatment and relevant factors in the control of parasite transmission and ongoing epidemiologic surveillance.

Reviewer #3: It could not be evaluated because of the appointments above described.

**Editorial and Data Presentation Modifications?**

Reviewer #1: (No Response)

Reviewer #2: Scientific names should be written in italic letters.

Reviewing the list of references.

Corrections are highlighted in yellow in the text. See attached manuscript.

Reviewer #3: Reject

**Summary and General Comments:**

Reviewer #1: Review PNTD-D-25-00569

General Comments: This is a very relevant manuscript, which updates and integrates a seroepidemiological survey with large-scale echocardiographic screening to estimate the prevalence of T. cruzi infection and identify associated risk factors in Feira de Santana, Bahia, Brasil. The manuscript is well written, presented, the methodology is in accordance with the objectives, and results. The main concern is the short introduction. A few other issues were also detected therefore, changes are recommended before the final acceptance of the manuscript in the PNTD.

Specific comments

Abstract please provide brief information on the place the study was carried out.

Introduction

Please provide a brief background on Bahia State related to epidemiological survey on Chagas disease as well as for Feira de Santana. The authors cite reference n. 4 however, we recommend a few words on the CD in Bahia and specially in Feira de Santana Municipality.

Also, it is recommended some words on the triatomine fauna in Bahia State, which presents the richest biodiversity of CD vector species, being one of the most infested states in Brasil. In addition to that new species have been described, revalidated in that state and also new triatomine species have been getting adapted to the human dwellings enhancing the risk of transmission of Chagas disease.

The authors mention the vectors as a risk factor for CD therefore, it is recommended including some references mentioning the vector species in Bahia.

Costa J, Almeida CE, Dotson EM, Lins A, Vinhaes M, Silveira AC, et al. The epidemiologic importance of Triatoma brasiliensis as a Chagas disease vector in Brazil: a revision of domiciliary captures during 1993-1999. Mem Inst Oswaldo Cruz. 2003; 98(4): 443-9.

Gurgel-Gonçalves R, Galvão C, Costa J, Peterson AT. Geographic distribution of Chagas disease vectors in Brazil based on ecological niche modeling. J Trop Med. 2012; 705326.

Costa, J.; Dale, C.; Galvão, C.; Almeida, C.E.; Dujardin, J.P. Do the new triatomine species pose new challenges or strategies for monitoring Chagas disease? An overview from 1979–2021. Mem. Inst. Oswaldo Cruz 2021, 116, e210015.

M & M

Please provide relevant information about Feira de Santana, physio geographic information based for instance in IBGE data and more specifically about the place the analyzed people reside, age, sex, socio economic status etc.

Discussion

From lines 312 – 315 please, include a reference in the text

From lines 333-335 please, include a reference in the text

Figs and Legends

Legends should be self-explanatory, please revise the legend of fig 1 providing information about the locality, the disease and any other relevant additional information. The same is applied for Table 1, fig 2 and Table 2

Reviewer #2: This study demonstrates how we can improve diagnostic strategies for T. cruzi infection using large-scale digital health AI tools aimed at overcoming barriers to identifying seropositive patients with excellent results. The significant association between household observation and the presence of triatomine highlights the ongoing exposure to the vector and the occurrence of parasite transmission in endemic areas. Given the possibility of increased risk of infection, it is important to emphasize the importance of early diagnosis using innovative methods, prompt etiologic treatment of cases, and well-designed clinical follow-up.

Reviewer #3: There are structural issues and a lack of clarity in the text. The methodology is not presented in full, and as a result, there is a lack of information to evaluate the results and discussion presented. The primary objective “estimate the seropositivity rate of T. cruzi infection” (lines 89-90) cannot be achieved with the proposal methodology.

PLOS authors have the option to publish the peer review history of their article (what does this mean? ). If published, this will include your full peer review and any attached files.

**Do you want your identity to be public for this peer review?** For information about this choice, including consent withdrawal, please see our Privacy Policy .

Reviewer #1: No

Reviewer #2: No

Reviewer #3: No

**Figure resubmission:**

**Reproducibility:**



---

## [Decision Letter · Decision Letter 1]

7 Dec 2025

Response to Reviewers
Revised Manuscript with Track Changes
Manuscript

Shaden Kamhawi

co-Editor-in-Chief

Paul Brindley

co-Editor-in-Chief

**Additional Editor Comments:** The revised manuscript represents a significant improvement, but we need an explicit consideration regarding the concern about "the absence of a second serological test to confirm infection" raised by Reviewer #3. Could the lack of such a confirmatory test have biased the results of the study?  ?>**Journal Requirements:**

**Reviewers' comments:**

**Key Review Criteria Required for Acceptance?**

**Methods**

-Are the objectives of the study clearly articulated with a clear testable hypothesis stated?

-Is the study design appropriate to address the stated objectives?

-Is the population clearly described and appropriate for the hypothesis being tested?

-Is the sample size sufficient to ensure adequate power to address the hypothesis being tested?

-Were correct statistical analysis used to support conclusions?

-Are there concerns about ethical or regulatory requirements being met?

Reviewer #1: The manuscript was greatly improved. The objectives are clear and the design is appropriate to address the objectives.

The manuscrilpt presents sufficient sample size to anwer the objectives. All concerns about ethical or regulatory requrements were clarified.

Reviewer #2: The objectives are clear and consistent with the cross-sectional study design. In this study, the sample population corresponds to the tested hypothesis, and the statistical analysis supports the conclusions. There are no ethical concerns.

Reviewer #3: The main point made by me in the previous version was not answered: the absence of a second serological test to confirm infection. This absence impacts the entire article because it lacks the necessary robustness to define those infected humans. Another unresolved point is associating the questionnaire on triatomine exposure only with serologically positive and not negative tests. The authors agreed and included this as a limitation, but simply including this limitation does not validate the analysis. Because I understand the first aspect impacts the accuracy of the diagnosis on which the entire article is based, I maintain my recommendation to reject the article.

**Results**

-Does the analysis presented match the analysis plan?

-Are the results clearly and completely presented?

-Are the figures (Tables, Images) of sufficient quality for clarity?

Reviewer #1: The results are clear and matcher the analysis plan and presented in full. Figs and tables are sufficient to illustrate the obtained data.

Reviewer #2: The data analysis corresponds to the study plan, which involved investigating the prevalence of Trypanosoma cruzi infection and associated risk factors in an area endemic for Chagas disease in the state of Bahia, Brazil using a cardiac screening programme. The seropositivity rate in this area is 8.5%. The data are clear and complete, figures and tables are self-explanatory and of a high standard.

Reviewer #3: (No Response)

**Conclusions**

-Are the conclusions supported by the data presented?

-Are the limitations of analysis clearly described?

-Do the authors discuss how these data can be helpful to advance our understanding of the topic under study?

-Is public health relevance addressed?

Reviewer #1: The conclusion is now addressing all relevant results obtained in the revised study encompassing a important topic on public heath once it updates the seroepidemiological and seropositivity data in Ferira de Santana, BA, Brazil. The obtained data was clearly discussed and contextualized in terms of advances and understanding Chagas disease in the studied area

Reviewer #2: The conclusions are appropriate to the data emphasizing the study limitations to study cross-sectional design. The study emphasises the significance of vector exposure in the transmission of Chagas disease, underscoring the necessity of targeted interventions. Further longitudinal studies involving larger community-based samples are required to investigate the temporal dynamics of infection risk by the parasite, and to validate these findings within broader populations.

Reviewer #3: (No Response)

**Editorial and Data Presentation Modifications?**

Reviewer #1: The manuscript can now be accepted for publication

Reviewer #2: I suggest reviewing the entire manuscript of the text and references.

Reviewer #3: (No Response)

**Summary and General Comments**

Reviewer #1: All issues were already addressed

Reviewer #2: I suggest that the authors review the entire manuscript, including the references. I recommend stating that transmission is of the etiological agent, T. cruzi, rather than of the disease itself, since most individuals with positive serology remain asymptomatic.

Reviewer #3: (No Response)

PLOS authors have the option to publish the peer review history of their article (what does this mean? ). If published, this will include your full peer review and any attached files.

**Do you want your identity to be public for this peer review?** For information about this choice, including consent withdrawal, please see our Privacy Policy .

Reviewer #1: No

Reviewer #2: No

Reviewer #3: No

**Figure resubmission:**

**Reproducibility:** To enhance the reproducibility of your results, we recommend that authors of applicable studies deposit laboratory protocols in protocols.io, where a protocol can be assigned its own identifier (DOI) such that it can be cited independently in the future. Additionally, PLOS ONE offers an option to publish peer-reviewed clinical study protocols. Read more information on sharing protocols at https://plos.org/protocols?utm_medium=editorial-email&utm_source=authorletters&utm_campaign=protocols

---

## [Editor Report · Decision Letter 2]

22 Dec 2025

Dear Dr Santos,

We are pleased to inform you that your manuscript 'Seroepidemiological survey and seropositivity rate for Trypanosoma cruzi infection in a community-based cardiac screening initiative in Feira de Santana, Bahia, Brazil' has been provisionally accepted for publication in PLOS Neglected Tropical Diseases.

Best regards,

Guilherme L Werneck

Section Editor

Guilherme Werneck

Section Editor

Shaden Kamhawi

co-Editor-in-Chief

Paul Brindley

co-Editor-in-Chief

---

## [Editor Report · Acceptance letter]

Dear Dr Santos,

We are delighted to inform you that your manuscript, " 

Seroepidemiological survey and seropositivity rate for Trypanosoma cruzi infection in a community-based cardiac screening initiative in Feira de Santana, Bahia, Brazil," has been formally accepted for publication in PLOS Neglected Tropical Diseases.

Best regards,

Shaden Kamhawi

co-Editor-in-Chief

Paul Brindley

co-Editor-in-Chief
